# ChatSR: Conversational Symbolic Regression

## Abstract

Formulas are the language of communication between humans and nature. It is an important research topic of artificial intelligence to find expressions from observed data to reflect the relationship between each variable in the data, which is called a symbolic regression problem. The existing symbolic regression methods directly generate expressions according to the given observation data, but we cannot require the algorithm to generate expressions that meet specific requirements according to the known prior knowledge. For example, the expression needs to contain the symbol 'sin' or be periodicity, and so on. Even if it can, it often requires very complex operations, which is very inconvenient. In this paper, based on multi-modal large language models, we propose ChatSR, a conversational symbolic regression method that can generate expressions that meet the requirements simply by describing the requirements with natural language instructions. By experimenting on the test datasets, we can demonstrate that ChatSR leads the state-of-the-art baselines in fitting performance. More notably, ChatSR can well understand the prior knowledge contained in natural language prompts, and can further improve the quality of generated expressions according to the prior knowledge. In addition, it is exciting that ChatSR has good zero-shot capability.

## 1 Introduction

Mathematical formulas are the language of communication between nature and human beings. With the succinct expression, we can obtain the potential relationship between the individual variables in the formula. The goal of scientists is to summarize a concise expression to reflect the law behind the physical phenomenon from the observation data. Because manual discovery of formula theorems often requires a long period. Moreover, the demands on scientists are high. So people are trying to use artificial intelligence algorithms to make computers discover mathematical formulas from data on their own. This is where the symbolic regression problem comes in. Specifically, given observations $D = \{X, y\}$, SR seeks a function $f$ that satisfies $y = f(X)$, where $X \in \mathbb{R}^{n \times d}$, $y \in \mathbb{R}^n$, $d, n$ are the dimension of variable and number of data points, respectively. $f$ is composed of several basic primitive operators such as $+, -, \times, \div, sin, cos, x_1, ....$

The traditional methods regard symbolic regression as a combinatorial optimization problem, and use GP algorithm and reinforcement learning algorithm to deal with symbolic regression problem. This kind of method is trained from scratch for each new data, which has good anti-noise and versatility.

However, in actual scientific research, scientists have a lot of priors or assumptions when modeling observation data with expressions. For example, suppose we use time t as a variable and want a function f(t) that models the intensity of light over a month. We almost know that f(t) is a periodic function, so we tend to model f(t) using periodic functions like sin() or cos(). In addition, scientists may want to obtain expressions that conform to symmetry, periodicity, translation invariance, and so on. Although current methods based on reinforcement learning can achieve some of the people's needs by adding certain constraints to the search process. But it will be very troublesome, requiring us to change the code, which is very unfriendly to non-computer science people.

In recent years, multimodal large language models (MLLMs), represented by GPT-4v, have achieved significant advancements. MLLMs can answer our questions based on input from various modalities such as images, PDF files, or videos. For example, if we input an image containing a person and a dog and then ask the MLLM, 'What are in the image?' it will describe the image by stating that 'There are a person and a dog'. If we add some conditions and ask, 'What pets is in the image?' it

will respond that 'There is a pet dog in the image'. So can we develop a multimodal large language model for symbolic regression, where we just take the input data [X,y], describe the requirements in natural langue, and it can generate the expression we need to fit the data [X,y]?

In this paper, we propose a symbolic regression method based on MLLM called ChatSR. ChatSR only needs to describe the requirements in simple natural language, and it can generate an expression that satisfies the requirements and fits the observed data. Specifically, we will consider the observed data as one modality and then consider the text (Question and answer pairs about expressions) as another modality. We first freeze the Large Language Model(LLM) and SetTransformer(Data Feature Extractor ) and then perform feature alignment by training a fully connected layer to map the observed data features to the literal feature space. In the second step, we unfreeze the parameters of LLM to train ChatSR End-to-End.

- This paper presents ChatSR, a conversational symbolic regression method based on multi-modal large language models, which allows describing requirements in natural language during chat interactions and generates expressions that meet those requirements.
- We find that trained ChatSR can leverage the powerful language understanding capabilities of large language models to have good zero-shot ability on properties or requirements outside the training set.
- We provide new potential research directions for using multimodal large language models.

## 2 RELATION WORK

### 2.1 MULTI-MODAL LARGE LANGUAGE MODELS

Recently, models such as CLIP Radford et al. (2021) and ALIGN Jia et al. (2021) have been pre-trained on noisy image and text pairs from the web using contrastive loss, which is recognized as one of the most effective methods for feature learning He et al. (2020)Chen et al. (2020)Li et al. (2020b)Li et al. (2020a). These models achieve remarkable performance on image-text retrieval tasks but are limited in their ability to model more complex interactions between images and text necessary for other vision-and-language (V+L) tasks Kim et al. (2021), such as visual question answering (VQA) Antol et al. (2015). Subsequent studies Wang et al. (2021) Wang et al. (2022a) Piergiovanni et al. (2022) have introduced encoder-decoder frameworks trained using generative loss functions, demonstrating robust performance across various vision-language benchmarks. Simultaneously, the visual encoders in these models maintain competitive accuracy in image classification tasks. Research Singh et al. (2022) Li et al. (2021) Li et al. (2022)Chen et al. (2023)Liu et al. (2024a) has explored the unification of image and text representations, which typically involves multiple pretraining stages for both unimodal and multimodal modules to achieve high performance. For instance, ALBEF Li et al. (2021) employs a dual-encoder architecture that integrates contrastive loss with Masked Language Modeling (MLM) to enhance learning efficiency. CoCa Yu et al. (2022) focuses on training an image-text foundation model from scratch in a single pretraining stage, thereby unifying these approaches in a simpler and more efficient manner. BEITv3 Wang et al. (2022b) treats images as a type of language, mapping images into the language space through a mapping layer before integrating them with encoded text features in a large GPT model. LLava Liu et al. (2024b) is an open-source multi-modal large model that aligns image features with text features to transform images into a 'language' that a large language model (LLM) Chang et al. (2023)Zhao et al. (2023)Touvron et al. (2023)Zeng et al. (2022)Ouyang et al. (2022) can understand. These image features are then concatenated with text features and fed into the large language model.

### 2.2 SYMBOLIC REGRESSION

**Based on genetic programming** This kind of method is a classical kind of algorithm in the field of symbolic regression. GP Arnaldo et al. (2014), McConaghy (2011), Nguyen et al. (2017) is the main representative of this kind of method, its main idea is to simulate the process of human evolution. Firstly, it initialized an expression population, then generated new individuals by crossover and mutation, and finally generated a new population by fitness. The above process is repeated until the target expression is obtained. RSRMXu et al. integrates the GP algorithm with Double Q-learningHasselt (2010) and the MCTS algorithmCoulom (2006). a Double Q-learning block, designed

for exploitation, that helps reduce the feasible search space of MCTS via properly understanding the distribution of reward, In short, the RSRM model consists of a three-step symbolic learning process: RLbased expression search, GP tuning, and MSDB. In this paperFong et al. (2022), the fitness function of the traditional GP algorithm is improved, which promotes the use of an adaptability framework in evolutionary SR which uses fitness functions that alternate across generations. LLM-SRChang et al. (2023) and ICSRZhao et al. (2023) use the large language model to aid the search process, just let the LLM produce a series of expressions, and then use the good expressions as a hint to let the LLM continue to produce a new batch of expressions until the target expression is reached.

**Based on reinforcement learning**    Reinforcement learning-based algorithms treat symbolic regression as a combinatorial optimization problem. The typical algorithm is DSRPetersen et al. (2019), which uses a recurrent neural network as a policy network to generate a probability distribution P for sampling, and then samples according to the probability P to obtain multiple expressions. The reward value of the sampled expressions is calculated and the policy network is updated with the risky policy, and the loop continues until the target expression is obtained. DSOMundhenk et al. (2021) is based on DSR by introducing the GP algorithm. The purpose of the policy network is to generate a better initial population for the GP algorithm. Then, the risk policy gradient algorithm is also used to update the policy network. Although the above two algorithms are very good, the efficiency is low, and the expression is more complex, especially the DSO algorithm is more obvious. There have been many recent symbolic regression algorithms based on the Monte Carlo tree search. SPLSun et al. (2022) uses MCTS in the field of symbolic regression and introduces the concept of modularity to improve search efficiency. However, due to the lack of guidance of MCTS, the search efficiency of this algorithm is low. To improve the search efficiency of the algorithm, the two algorithms DGSR-MCTS Kamienny et al. (2023) and TPSR Shojaee et al. (2024a) introduced the policy network to guide the MCTS process based on the previous algorithm. While maintaining the performance of the algorithm, it greatly improves the search efficiency of the algorithm. However, although the above two algorithms improve the search efficiency of the algorithm, they reduce the Versatility of the algorithm, and the noise robustness ability of the algorithm is also greatly reduced. To solve the above problems and balance the Versatility and efficiency of the algorithm, SR-GPT Li et al. (2024a) uses a policy network that learns in real-time to guide the MCTS process. It achieves high performance while efficient search.

**Based on pre-training**    Many SR methods based on reinforcement learning have good Versatility. However, its search efficiency is relatively low, and it often takes a long time to get a good expression. In contrast, pre-trained models treat the SR problem as a translation problem and train a transformer with a large amount of artificially synthesized data in advance. Each prediction only needs one forward propagation to get the result, which is relatively efficient. SymbolicGPTValipour et al. (2021) was the first large-scale pre-trained model to treat each letter in a sequence of symbols as a token, (e.g.['s',' i','n', '(', 'x', ')']). A data feature extractor is used as the encoder, and then each token is generated by the Decoder in turn. Finally, the predicted sequence and the real sequence are used for cross-entropy loss. BFGS is used to optimize the constant at placeholder 'C'. NeSymReSBiggio et al. (2021) builds on symbolicGPT by not thinking of each individual letter in the sequence of expressions as a token. Instead, Nesymres represents the expression in the form of a binary tree, which is then expanded by preorder traversal, and considers each operator as a token (e.g. ['sin','x']). Then SetTransformer is used as the Encoder of the data, and finally, Decoder is used to generate the expression sequence. The overall framework and idea of the EndtoEndKamienny et al. (2022) algorithm are not much different from NeSymReS, but EndtoEnd abandons the constant placeholder 'C', encodes the constant, and directly generates the constant from the decoder. The constants are then further optimized by Broyden-Fletcher-Goldfarb-Shanno (BFGS) Liu & Nocedal (1989). Based on EndtoEnd, NSRwHBendinelli et al. (2023) tries to apply some prefixes to prompt the model to generate expressions that conform to the prior. But the effect is not obvious. SymformerVastl et al. (2024) is slightly different from the previous pre-trained models in that it directly generates the constant values in the expression as well as the sequence of expressions. LLM-SRShojaee et al. (2024b) and ICSRMerler et al. (2024) use LLM as a guide, but do not train LLM specifically for SR tasks. SNIPMeidani et al. (2023) first applies contrastive learning to train the feature encoder and then freezes the encoder to train the decoder. But SNIP works well only when combined with a latent space optimization (LSO)Bojanowski et al. (2017) algorithm. MMSRLi et al. (2024b) solves the symbolic regression problem as a pure multimodal problem, takes the input data and the expression

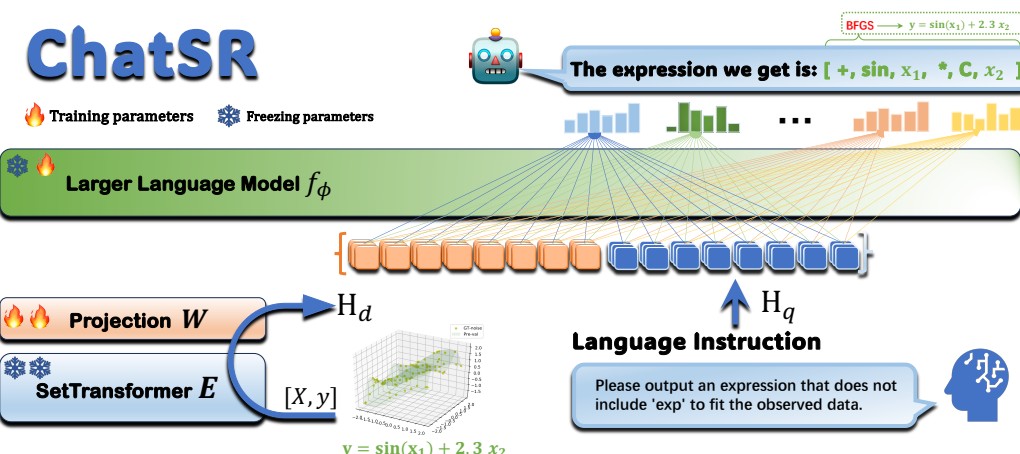

Figure 1: This figure shows a schematic diagram of the overall process of ChatSR.

sequence as two modalities, introduces contrastive learning in the training process, and adopts a one-step training strategy to train contrastive learning with other losses.

**Based on deep learning** This class of methods combines symbolic regression problems with artificial neural networks, where EQL replaces the activation function in ordinary neural networks with [sin, cos,...] And then applies pruning methods to remove redundant connections and extract an expression from the network. EQLKim et al. (2020) is very powerful, however, it can't introduce division operations, which can lead to vanishing or exploding gradients. The main idea of AI Feynman 1.0 Udrescu & Tegmark (2020) and AI Feynman 2.0Udrescu & Tegmark (2020) series algorithms is to "Break down the complex into the simple" by first fitting the data with a neural network, and then using the trained neural network to discover some properties (e.g. Symmetry, translation invariance, etc.) to decompose the function hierarchically. AI Feynman 2.0 introduces more properties based on AI Feynman 1.0, which makes the scope of its application more extensive relative to AI Feynman 1.0. MetaSymNetLi et al. (2023) takes advantage of the differences between symbolic regression and traditional combinatorial optimization problems and uses more efficient numerical optimization to solve symbolic regression.

## 3 METHOD

We manually generate 15M Q&A data about expressions. Each piece of data contains a set of observations, [X, Y], and a question-answer pair. Where the problem contains our requirements for generating expressions. (e.g., sin() free, symmetry, etc.). The answer mainly consists of a preorder traversal of the generates expression. We first train a SetTransformer as the data feature extractor E of ChatSR using contrastive learning with 1M pairs of [X, Y] and the corresponding expression preorder traversal (e.g.$[sin, *, x, x]$)Meidani et al. (2023).Its training process is shown in Fig.2 Then, we freeze the parameters of SetTransformer and LLM and separately pre-train the parameters of the projection layer to map data features to word features. Finally, only the parameters SetTransformer for are frozen, and the parameters of the projection layer and LLM are trained. The flow chart of the ChatSR is shown in Fig.1. Note the parameters of the LLM we trained with LoRAHu et al. (2021).

### 3.1 EXPRESSIONS GENERATION

In ChatSR, we use symbols $[+, -, *, /, sin, cos, log, sqrt, C, x_1, x_2, ..., x_n]$. Here, $C$ denotes a constant placeholder (for example, sin(2.6x) can be written as sin(C*x), with its preorder traversal of expression binary tree being [sin, *, C, x]), and $[x_1, ..., x_n]$ represents variables. Expressions composed of these symbols can be represented in the form of a binary tree. Then,by performing a preorder traversal of the binary tree, we can obtain a sequence of symbols. Therefore, we randomly

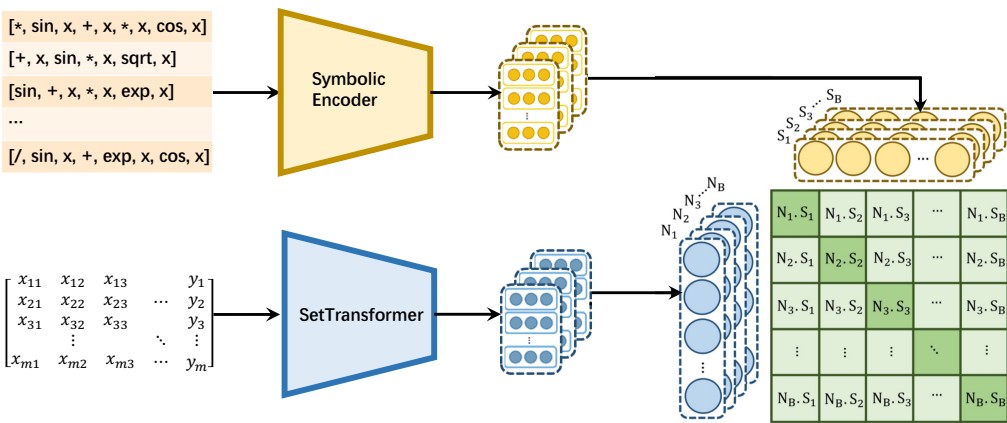

Figure 2: The feature extractor Setransformer is pretrain by contrastive learning.

sample a sequence of symbols in the above symbol library in some rule3.1.1. The sequence of symbols is then restored to an expression and sampled to obtain [X, Y].

### 3.1.1 GENERATION STOP DECISION: $count = 0$

To determine whether the expression generation should be stopped, we import a counter variable, $count$, and initialize it to 1. In addation, we introduce the Arity($s$) function, if $s$ is a binary operator, e.g. $[+, -, *, /]$, then Arity($s$)=2; Similarly, if $s$ is a unary operator, e.g. $[sin, cos, ...]$, then Arity($s$)=1; If $s$ is a variable $[x_1, ..., x_n]$ or a constant placeholder 'C', then Arity($s$)=0. First, we randomly select a symbol, $s$, from the symbol library and update the $count$ according to the formula $count = count - Arity(s) - 1$. This process is repeated until $count = 0$. At this point, we have constructed a complete sequence of expressions.

### 3.1.2 GENERATION CONSTRAINTS

To ensure the generated expressions are meaningful, we impose the following restrictions:
1. Trigonometric functions cannot be nested (e.g., $\sin(\cos(x))$), as such forms are rarely encountered in real-life scenarios.
2. For functions like $\log(x)$ and $\sqrt{x}$, the argument $x$ cannot be negative. For instance, expressions like $\log(\sin(x))$ and $\sqrt{\cos(x)}$ are invalid because both $\sin(x)$ and $\cos(x)$ can take negative values.

### 3.2 CONSTANT OPTIMIZATION

LLM will first generate the preorder traversal of the expression, and then for the expression with constant C, we will use function-calling technology to call BFGS and other numerical optimization algorithms to optimize the constant. Finally, we will output the optimized expression with a specific value. For example, if we get a preorder traversal of an expression, $[*, C, sin, x]$, the corresponding expression is $C * sin(x)$, then we need to use the BFGS algorithm to optimize the constant value at C with X as input and y as output.

### 3.3 TRAINING DATA COLLECTION

In section3.1, we generated a number of expressions, and for each expression, we can generate multiple $Q\&A$ text data according to certain rules. For each expression, we have the expression preorder traversal, various properties (periodicity, symmetry), and preorder traversal length. With this information, we can generate question-answering data like the one in Fig.3. To be specific, For each piece of data. We generate question-answer pairs by the following method. Specifically, we have a <Data> symbol in front of each request sentence, indicating that this is a data feature.

# Data details

**Expression:** y= $2\sin(x_1)\cos(x_2)$

**Preorder traversal :** $[*,\ c,\ *,\ sin,\ x_1,\ cos, x_2]$

**Properties :** Periodicity, symmetry

**Number of symbols :** 7

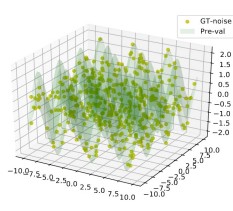

$\{X, Y\}$

# Conversations

### Conversation 1

**Human:** <Data> Generate an expression to fit the data for me based on the uploaded data.
**Assistant:** Of course, according to your requirements, the expression I generated is $[*,\ c,\ *,\ sin,\ x_1,\ cos, x_2]$.

### Conversation 2

**Human:** : <Data> I want to expression with periodicity on the variable x1 to fit the above data.
**Assistant:** According to your requirements, we will generate the expression for you is $[*,\ c,\ *,\ sin,\ x_1,\ cos, x_2]$.

### Conversation 3

**Human:** : <Data> I want to generate an expression with symbols [sin, cos] to fit the above data.
**Assistant:** Well, we get an expression for you is $[*,\ c,\ *,\ sin,\ x_1,\ cos, x_2]$.

### Conversation 4

**Human:** : <Data> I want to generate an expression with 10 symbols or less to fit the above data.
**Assistant:** OK, the expression I found is $[*,\ c,\ *,\ sin,\ x_1,\ cos, x_2]$.

### Conversation 5

**Human:** : <Data> I want to generate an expression consisting of the symbols $[*, sin, cos, x_1, x_2]$ to fit the above data.
**Assistant:** We generate the expression for you is $[*,\ c,\ *,\ sin,\ x_1,\ cos, x_2]$.

### Conversation 6

**Human:** : <Data> Generate an expression to fit the data. Note that the data is noisy, so be careful about robustness.
**Assistant:** We generate the expression for you is $[*,\ c,\ *,\ sin,\ x_1,\ cos, x_2]$.

Figure 3: Data details and templates. Schematic representation of the expression $y = 2\sin(x_1)\cos(x_1)$ related question-answer pair generation.

- 1, we generate a question-answer pair that only asks the model to generate expressions that fit the data, as in traditional symbolic regression.

- 2, generate expressions that satisfy some property, such as the expression $y = 2sin(x_1)cos(x_2)$, which is periodic with respect to both variables. We can then ask the model to generate expressions that are periodic with respect to $x_1$, periodic with respect to $x_2$, and periodic with respect to each variable. The same goes for symmetries.

- 3, In order to require that certain symbols must be included in the expressions generated by ChatSR, we randomly choose $k$ symbols at a time from the preorder traversal S, where $k < len(S)$, is a random integer between 1 and len(S). Then, the selected symbols are embedded into the corresponding sentence to obtain a complete sentence.

- 4, We require the length of the preorder traversal of the expression generated by ChatSR to be less than a certain number $M_L$. We will add a random integer between 0 and 20 to the length of the expression preorder traversal to obtain the number $M_L$. Then $M_L$ is embedded into the corresponding statement to obtain a complete dialogue.

- 5, We want ChatSR to generate expressions using only certain symbols. We will process the preorder traversal of the expression to filter out repeated symbols and then embed the resulting sequence of symbols into the corresponding statement to obtain a complete dialogue.

- 6, If we want ChatSR to have some noise immunity, we randomly add some Gaussian noise with different levels to the clean data. We then generate question-answer pairs about generating robust expressions from noisy data.

## 3.4 MODEL ARCHITECTURE

In ChatSR1, we use SetTransformer trained with contrastive learning as the data feature extractor, whose parameters are frozen throughout. Then the data features extracted by SetTransformer are mapped to the word embedding space of LLM through the projection layer. Finally, we train the parameters of the projection layer and LLM together. We choose VicunaChiang et al. (2023) as our LLM $f_\varphi$ parameterized by $\varphi$.

### 3.4.1 SETTRANSFORMER

The data information plays a crucial role in guiding the Decoder. To accommodate the permutation invariance of data features—where the dataset's features should remain unchanged regardless of the input order—we utilize the SetTransformer as our data encoding method, as described by Lee et al. (2019). Our encoder takes a set of data points $\mathcal{D} = \{X, y\} \in \mathbb{R}^{n \times d}$. These data points undergo an initial transformation via a trainable affine layer, which uplifts them into a latent space $h_n \in \mathbb{R}^{d_h}$. Subsequently, the data is processed through a series of Induced Set Attention Blocks (ISABs)Lee et al. (2019), which employ several layers of cross-attention mechanisms. Initially, a set of learnable vectors serves as queries, with the input data acting as the keys and values for the first cross-attention layer. The outputs from this first layer are then repurposed as keys and values for a subsequent cross-attention process, with the original dataset vectors as queries. Following these layers of cross-attention, we introduce a dropout layer to prevent overfitting. Finally, the output size is standardized through a final cross-attention operation that uses another set of learnable vectors as queries, ensuring that the output size remains consistent and does not vary with the number of inputs.

### 3.5 MODEL TRAINING

For each piece of data $X_D = [X, y]$, we have multi-turn question and answer pairs $[X_q^1, X_a^1, X_q^2, X_a^2, ..., X_q^T, X_a^T]$. Where T stands for the total turns of question-answering. We organize them as a sequence $[X_D, X_q^1, X_a^1, X_q^2, X_a^2, ..., X_q^T, X_a^T]$ by treating all answers as the assistant's response, and the instruction $X_{\text{instruct}}^t$ at the $t^{\text{th}}$ turn as follows:

$$X_{\text{instruct}}^t = \begin{cases} [X_D, X_q^1], & \text{When t = 1} \\ X_q^t, & \text{When t > 1} \end{cases} \tag{1}$$

We perform instruction-tuning of the LLM on the prediction tokens, using its original auto-regressive training objective. Specifically, for a sequence of length L, we compute the probability of the target answers $X_a$ by:

$$p(X_a \mid X_D, X_{\text{instruct}}) = \prod_{i=1}^{L} p_\theta(x_i \mid X_D, X_{\text{instruct}, <i}, X_{a, <i}) \tag{2}$$

where $\theta$ is the trainable parameters, $X_{instruct, <i}$ and $X_{a, <i}$ are the instruction and answer tokens in all turns before the current prediction token $x_i$, respectively. We explicitly add $X_D$ to emphasize the fact that the data is grounded for all answers. For ChatSR model training, we consider a two-stage instruction-tuning procedure.

**Stage 1: Pre-training for Feature Alignment.** In the first training step, we take 600K samples from all the datasets (including $[X_D, X_q, X_a]$) for feature alignment training. During training, we keep both the SetTransformer and LLM weights frozen and maximize the likelihood of Eq.2 using

only the trainable parameters $\theta = W$ (the projection matrix). This allows the data features $H_v$ to be aligned with the pre-trained LLM word embeddings. This stage can be understood as training a compatible data tokenizer for the frozen LLM.

**Stage 2: Fine-tuning End-to-End.** We always keep the SetTransformer weights frozen, and continue to update both the pre-trained weights of the projection layer and LLM in LLaVA; (the trainable parameters are $\theta = \{W, \varphi\}$ in Fig.1).

| Group | Dataset | ChatSR | | MMSR | | SNIP | | NeSymRes | | TPSR | |
|---|---|---|---|---|---|---|---|---|---|---|---|
| | | $R^2 \uparrow$ | Nodes $\downarrow$ | $R^2 \uparrow$ | Nodes $\downarrow$ | $R^2 \uparrow$ | Nodes $\downarrow$ | $R^2 \uparrow$ | Nodes $\downarrow$ | $R^2 \uparrow$ | Nodes $\downarrow$ |
| Standards | Nguyen | **0.9999**$_{\pm 0.001}$ | **12.1** | **0.9999**$_{\pm 0.001}$ | 14.5 | 0.9936$_{\pm 0.004}$ | 17.1 | 0.8568$_{\pm 0.003}$ | 18.2 | 0.9942$_{\pm 0.005}$ | 35.2 |
| | Keijzer | **0.9992**$_{\pm 0.003}$ | **13.4** | 0.9924$_{\pm 0.003}$ | 16.3 | 0.9862$_{\pm 0.005}$ | 20.4 | 0.7992$_{\pm 0.003}$ | 21.3 | 0.9819$_{\pm 0.004}$ | 36.4 |
| | Korns | **0.9941**$_{\pm 0.003}$ | **16.4** | 0.9927$_{\pm 0.003}$ | 19.2 | 0.9418$_{\pm 0.004}$ | 21.9 | 0.8011$_{\pm 0.005}$ | 23.5 | 0.9288$_{\pm 0.005}$ | 39.4 |
| | Constant | 0.9925$_{\pm 0.002}$ | 20.5 | **0.9946**$_{\pm 0.002}$ | 24.5 | 0.9299$_{\pm 0.003}$ | 23.4 | 0.8344$_{\pm 0.003}$ | 24.1 | 0.9344$_{\pm 0.004}$ | 42.1 |
| | Livermore | **0.9885**$_{\pm 0.003}$ | 23.6 | 0.9726$_{\pm 0.003}$ | 29.4 | 0.8948$_{\pm 0.004}$ | 34.6 | 0.6836$_{\pm 0.005}$ | 2.9 | 0.8828$_{\pm 0.005}$ | 56.3 |
| | Vladislavleva | **0.9884**$_{\pm 0.003}$ | **16.8** | 0.9812$_{\pm 0.003}$ | 21.7 | 0.9212$_{\pm 0.005}$ | 39.4 | 0.6892$_{\pm 0.004}$ | 36.2 | 0.9433$_{\pm 0.004}$ | 69.8 |
| | R | **0.9948**$_{\pm 0.004}$ | **14.3** | 0.9811$_{\pm 0.004}$ | 16.4 | 0.9614$_{\pm 0.004}$ | 24.8 | 0.7703$_{\pm 0.005}$ | 27.3 | 0.9529$_{\pm 0.005}$ | 47.2 |
| | Jin | **0.9962**$_{\pm 0.003}$ | 21.6 | 0.9902$_{\pm 0.003}$ | 28.3 | 0.9877$_{\pm 0.004}$ | 17.2 | 0.8327$_{\pm 0.003}$ | 19.9 | 0.9626$_{\pm 0.006}$ | 43.3 |
| | Neat | 0.9943$_{\pm 0.004}$ | **12.7** | **0.9952**$_{\pm 0.004}$ | 17.3 | 0.9401$_{\pm 0.004}$ | 18.9 | 0.7596$_{\pm 0.005}$ | 20.6 | 0.9528$_{\pm 0.005}$ | 38.2 |
| | Others | 0.9936$_{\pm 0.002}$ | 15.3 | **0.9968**$_{\pm 0.002}$ | 20.6 | 0.9702$_{\pm 0.003}$ | 30.4 | 0.8026$_{\pm 0.003}$ | 32.2 | 0.9625$_{\pm 0.004}$ | 48.5 |
| SRBench | Feynman | **0.9910**$_{\pm 0.002}$ | **16.4** | 0.9874$_{\pm 0.002}$ | 20.8 | 0.8899$_{\pm 0.004}$ | 21.1 | 0.7025$_{\pm 0.005}$ | 22.4 | 0.9184$_{\pm 0.005}$ | 45.1 |
| | Strogatz | **0.9861**$_{\pm 0.003}$ | 28.7 | 0.9819$_{\pm 0.003}$ | 39.6 | 0.8307$_{\pm 0.003}$ | 26.8 | 0.6022$_{\pm 0.003}$ | 28.1 | 0.8611$_{\pm 0.004}$ | 32.3 |
| | Black-box | 0.8921$_{\pm 0.004}$ | **18.9** | **0.9037**$_{\pm 0.004}$ | 26.7 | 0.8692$_{\pm 0.004}$ | 29.2 | 0.6525$_{\pm 0.005}$ | 33.9 | 0.9024$_{\pm 0.005}$ | 54.2 |
| | Average | **0.9854** | **17.7** | 0.9820 | 21.3 | 0.9321 | 25.0 | 0.7528 | 26.2 | 0.9368 | 45.2 |

Table 1: The results of performance comparison. At a 0.95 confidence level, a comparison of the coefficient of determination ($R^2$) and the expression complexity(Nodes) was conducted between ChatSR and four baselines.

## 4 EXPERIMENT

To verify the performance of the ChatSR algorithm, we tested it on the 13 datasets. We selected four state-of-the-art baselines to compare with ChatSR. The details of the four baselines are as follows:

- **MMSRLi et al. (2024b)**. A pre-training method that treats symbolic regression as a multi-modal problem and uses contrastive learning for modal alignment.
- **TPSRShojaee et al. (2024a)**. A symbolic Regression algorithm combining large-scale pre-trained models and Monte Carlo Tree Search.
- **NeSymReSBiggio et al. (2021)**. This algorithm is categorized as a large-scale pre-training model.
- **SNIPMeidani et al. (2023)**. A large-scale pre-trained model with a feature extractor trained with contrastive learning before training.

### 4.1 COMPARISON WITH BASELINES

#### 4.1.1 COMPARISON OF $R^2$

The most important goal of symbolic regression is to find an expression from the observed data that accurately fits the given data. A very important indicator to judge the goodness of fit is the coefficient of determination ($R^2$). Therefore, we tested the five algorithms on 13 datasets (Detail in AppendixD,C), using $R^2$ as the standard. We run each expression in the dataset 20 times and then take the average $R^2$ of all the expressions in the dataset. And the confidence levelJunk (1999), Costermans et al. (1992) is taken to be 0.95. The specific results are shown in Table 1($R^2$).

#### 4.1.2 COMPARISON OF RECOVER RATE

Recovery rate is also a more stringent metric than R2. It is a test of how well an algorithm can fully recover an expresse.R2 must be equal to or have the potential to be equal to 1.0. For example, for the expression sin(x), sin(x)+c and sin(c*x) both count as full recovery. Specifically, for each dataset, we count the ratio of the number of times the expression is fully recovered across all tests and the total number of tests. The detailed average recovery of each dataset is shown in Fig.4.

### 4.1.3 COMPARISON OF COMPLEXITY(NODES OF EXPRESSION)

The complexity of the expression (the number of nodes) is also an important index to evaluate symbolic regression algorithms. Because for symbolic regression, it doesn't make sense to apply a very complex expression to fit the data. In this experiment, for each algorithm, we tested each expression 20 times, and then recorded the average R2 and the average expression complexity. The results are shown in Table1 (Nodes).

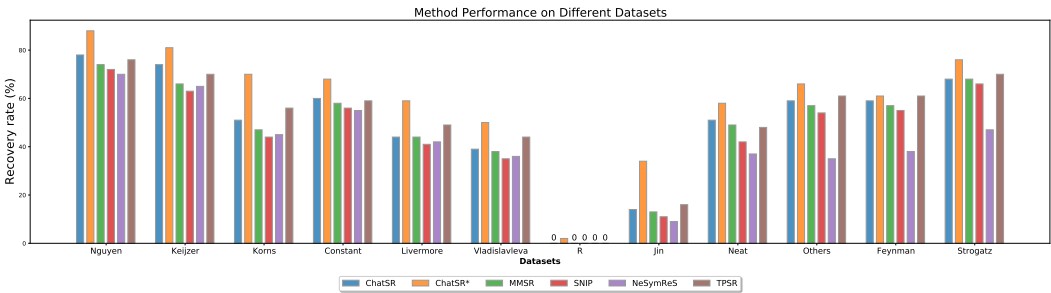

Figure 4: Recover rate of various algorithms. Note: ChatSR's prompt does not contain prior knowledge. However, ChatSR*'s prompt carries prior knowledge. From the figure, we can see that introducing prior knowledge into the instruction can effectively improve the recover rate.

| Points | Monotonically increasing | | Monotonically decreasing | | central symmetry | | convexity | | concavity | |
|---|---|---|---|---|---|---|---|---|---|---|
| Prior-k (Use/No) | Use Prior-k | No Prior-k | Use Prior-k | No Prior-k | Use Prior-k | No Prior-k | Use Prior-k | No Prior-k | Use Prior-k | No Prior-k |
| $R^2$ | 0.9992 | 0.9936 | 0.9993 | 0.9902 | 0.9953 | 0.9885 | 0.9999 | 0.9905 | 0.9998 | 0.9902 |
| Recover rate | 88.6% | 56.2% | 83.0% | 52.7% | 73.4% | 44.9% | 80.5% | 48.3% | 76.0% | 46.7% |
| Success rate | 96.2% | 66.2% | 97.0% | 62.2% | 90.4% | 38.8% | 98.5% | 88.2% | 97.0% | 86.2% |

Table 2: The Zero-shot proficiency test. In the table, 'Prior-k' denotes 'Prior knowledge'. Here, 'Use prior-k' and 'No prior-k' denote whether Prior knowledge of the relevant properties is introduced in the prompt, respectively. The 'Success rate' represents the proportion of generated expressions that conform to the corresponding property.

### 4.1.4 ANALYSIS OF RESULT

From the above results we can see that although ChatSR is only slightly ahead of MMSR in average $R^2$. However, ChatSR is significantly better than the others in terms of expression complexity, which we think may be due to the fact that large language models already know that "the more concise the expression, the better". As a result, ChatSR produces much more concise expressions than MMSR and the other baselines. It is worth noting that on the more difficult recovery rate, thanks to the powerful knowledge reserve of the large language model, ChatSR significantly outperforms the other algorithms after introducing prior knowledge in the prompt.

### 4.2 ABLATION EXPERIMENT: THE EFFECT OF PRIOR KNOWLEDGE ON THE RESULTS

Expression recovery rate is a very challenging metric to evaluate symbolic regression algorithms. In order to test whether introducing prior knowledge into the prompts improves ChatSR's recovery rate, we tested them on the datasets. First, in the prompt, we don't give it any prior knowledge (no requirements), we just ask it to generate an expression to fit the data. Then, we give the model some prior knowledge (requirements) in the prompt, according to the properties and form of the expressions . For example, for Nguyen-5 $\sin(x^2)\cos(x) - 1$, we'll ask it to generate an expression that contains the symbols $\sin$ and $\cos$.

For each expression, we do the above two kinds of experiments 20 times each. Finally, the recovery rate was calculated. The specific results are shown in Fig.4(ChatSR, no prior knowledge in prompt ) and ChatSR*, giving prior knowledge in prompt). From the figure, we can clearly see that giving prior knowledge of the prompt can significantly improve the recovery rate. This also proves that our aim of improving the quality of generated expressions by providing priors through natural language

is achieved. More importantly, ChatSR does have the ability to generate expressions that match our needs.

### 4.3 ZERO-SHOT ABILITY TEST

The general language model has a strong natural language understanding ability and a rich knowledge reserve. Previous experiments have found that it has a strong zero-shot ability on many tasks. Therefore, we would like ChatSR to inherit this capability as well. For example, if we only included symmetries and periodicity in our training dataset, could ChatSR use the zero-shot power of large language models to directly understand and generate expressions with other properties? (e.g., the generated expression should be monotonic, etc.).

To test ChatSR's zero-shot capability, we chose several properties not covered in the training dataset to test (monotonicity, symmetry with respect to the origin, convexity, concavity, and boundedness). The main test is whether ChatSR can understand and generate expressions that meet the requirements according to its zero-shot ability. Specifically, for each property above, we artificially synthesize 10 expressions(Appendix B) that conform to this property. Then we run each expression 20 times with different prompts (whether we want the generated expression to satisfy a certain property or not). The averaged $R^2$, recover rate and the proportion (Success rate) of generated expressions conforming to the specified properties are then counted. The test results are shown in Table 2. As we can see, ChatSR shows good zero-shot ability. When prompted, ChatSR was able to generate expressions that matched the requirements, even if those properties were not present in the training dataset. And $R^2$ and recover rate are also improved. This is due to the powerful language understanding capabilities of large language models.

## 5 CONCLUSION AND DISCUSSION

In this work, we present ChatSR, a novel symbolic regression paradigm based on multimodal large language models that can conduct conversations and make requests in natural language. Specifically, we use many data and dialog pairs to train a multimodal large language model with the ability to fit the data by generating expressions based on the data and natural language prompts. Moreover, we experimentally find that ChatSR has good zero-shot capability. This means that we can rely on the powerful understanding power of large language models to impose various requirements on expression generation. This approach promises to change the way symbolic regression can be applied. When we want the generated expression to satisfy some constraint, we only need to describe our requirements in natural language instead of changing the code. This greatly reduces the threshold for the use of symbolic regression and improves the flexibility of the symbolic regression algorithm. ChatSR will have great potential applications in finance, healthcare, and other fields that have very high requirements for interpretability because ChatSR can get an interpretable mathematical expression from the data. In addition, we believe ChatSR has great potential for applications in scientific discovery and AI for Science.

Last but not least, ChatSR also has some problems, such as poor noise robustness. Next, we will try to improve its noise robustness by contrastive learning or other methods.

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

# APPENDIX FOR "CHATSR: CONVERSATIONAL SYMBOLIC REGRESSION"

## A APPENDIX: DETAILED SETTINGS OF HYPERPARAMETERS DURING TRAINING THE SETTRANSFORMER.

Table 3: Hyperparameters of SetTransformer

| hyperparameters | Numerical value |
|---|---|
| N_p | 0 |
| activation | 'relu' |
| bit16 | True |
| dec_layers | 5 |
| dec_pf_dim | 512 |
| dim_hidden | 512 |
| dim_input | 3 |
| dropout | 0 |
| input_normalization | False |
| length_eq | 60 |
| linear | False |
| ln | True |
| lr | 0.0001 |
| mean | 0.5 |
| n_l_enc | 5 |
| norm | True |
| num_features | 20 |
| num_heads | 8 |
| num_inds | 50 |
| output_dim | 60 |
| sinuisodal_embeddings | False |
| src_pad_idx | 0 |
| std | 0.5 |
| trg_pad_idx | 0 |

## B  APPENDIX: DETAILS OF THE EXPRESSIONS FOR THE VARIOUS PROPERTIES INVOLVED IN THE ZERO-SHOT EXPERIMENT

To test ChatSR's Zero-shot capability, we designed some properties not included in the training dataset for testing. They include Continuous Monotonic Decreasing, Continuous Globally Monotonically Increasing, Origin-Centered Symmetric, Continuous Convex and Continuous Concave are the five properties. For each property, 10 functions satisfying this property are designed. The form of the function is as follows Table 4,5,6,7,8.

| Function Index | Expression | Domain |
|---|---|---|
| 1 | $f(x) = -x - \ln(x+1)$ | $x \geq 0$ |
| 2 | $f(x) = e^{-x} - x^2$ | $\mathbb{R}$ |
| 3 | $f(x) = \frac{1}{x+1} - \sqrt{x}$ | $x > 0$ |
| 4 | $f(x) = 10 - x^2 - \arctan(x)$ | $\mathbb{R}$ |
| 5 | $f(x) = \frac{1}{\sqrt{x+1}} - \ln(x+2)$ | $x \geq 0$ |
| 6 | $f(x) = e^{-x}\cos(x) + \frac{1}{x+1}$ | $x > 0$ |
| 7 | $f(x) = -\ln(x+1) + x^{-0.5}$ | $x > 0$ |
| 8 | $f(x) = \sqrt{x+1} - 3\ln(x+2)$ | $x \geq 0$ |
| 9 | $f(x) = e^{-x^2} - x$ | $\mathbb{R}$ |
| 10 | $f(x) = -x^{3/2} - \tan^{-1}(x)$ | $x \geq 0$ |

Table 4: List of Continuous Monotonic Decreasing Functions

| Function Index | Expression | Domain |
|---|---|---|
| 1 | $f(x) = x + \ln(x^2 + 1)$ | $\mathbb{R}$ |
| 2 | $f(x) = e^x + x^2$ | $\mathbb{R}$ |
| 3 | $f(x) = x + \arctan(x)$ | $\mathbb{R}$ |
| 4 | $f(x) = x\sqrt{x^2 + 1}$ | $\mathbb{R}$ |
| 5 | $f(x) = x^3 + 3x$ | $\mathbb{R}$ |
| 6 | $f(x) = x + \sqrt{x+2} + \ln(x^2 + 1)$ | $x \geq -2$ |
| 7 | $f(x) = e^x + \ln(x^2 + 1)$ | $\mathbb{R}$ |
| 8 | $f(x) = \sqrt{x^2 + 1} + x^3$ | $\mathbb{R}$ |
| 9 | $f(x) = x + \arcsin(\tanh(x))$ | $\mathbb{R}$ |
| 10 | $f(x) = \ln(x+2) + e^x$ | $x > -2$ |

Table 5: List of Continuous Globally Monotonically Increasing Functions

| Function Index | Expressions | Domain |
|---|---|---|
| 1 | $f(x) = x \sin(x)$ | $x \in \mathbb{R}$ |
| 2 | $f(x) = 3x^3 - 2x$ | $x \in \mathbb{R}$ |
| 3 | $f(x) = \log(1+x) - \log(1-x)$ | $x \in (-1, 1)$ |
| 4 | $f(x) = e^x - e^{-x}$ | $x \in \mathbb{R}$ |
| 5 | $f(x) = \arctan(x) - \arctan(-x)$ | $x \in \mathbb{R}$ |
| 6 | $f(x) = x(x^2 + 3)$ | $x \in \mathbb{R}$ |
| 7 | $f(x) = x^5 - 10x^3 + 9x$ | $x \in \mathbb{R}$ |
| 8 | $f(x) = \sinh(x) = \frac{e^x - e^{-x}}{2}$ | $x \in \mathbb{R}$ |
| 9 | $f(x) = 7x - x^7$ | $x \in \mathbb{R}$ |
| 10 | $f(x) = x \cos(x) + \sin(x)$ | $x \in \mathbb{R}$ |

Table 6: Expressions that are Origin-Centered Symmetric (Odd Functions)

| Function Index | Expression | Domain |
|---|---|---|
| 1 | $f(x) = x^4 + 2x^2 + 1$ | $x \in \mathbb{R}$ |
| 2 | $f(x) = e^x + x^2$ | $\mathbb{R}$ |
| 3 | $f(x) = x^2 + \log(1 + e^x)$ | $x \in \mathbb{R}$ |
| 4 | $f(x) = x \sinh(x) + \cosh(x)$ | $x \in \mathbb{R}$ |
| 5 | $f(x) = x^2 + \sqrt{x^2 + 1}$ | $x \in \mathbb{R}$ |
| 6 | $f(x) = e^{x^2} - x$ | $x \in \mathbb{R}$ |
| 7 | $f(x) = x^2 + \arctan(x)$ | $x \in \mathbb{R}$ |
| 8 | $f(x) = \sqrt{1 + e^x}$ | $x \in \mathbb{R}$ |
| 9 | $f(x) = x + \log(1 + x^2)$ | $x \in \mathbb{R}$ |
| 10 | $f(x) = x^6 + x^4 - x^3 + x + e^{-x}$ | $x \in \mathbb{R}$ |

Table 7: List of Continuous Convex Functions

| Function Index | Expression | Domain |
|---|---|---|
| 1 | $f(x) = -x^4 + 2x^2 + 1$ | $x \in \mathbb{R}$ |
| 2 | $f(x) = -e^x + 3x - 2$ | $x \in \mathbb{R}$ |
| 3 | $f(x) = -x^2 + \log(1 + x^2)$ | $x \in \mathbb{R}$ |
| 4 | $f(x) = -\cosh(x)$ | $x \in \mathbb{R}$ |
| 5 | $f(x) = -x^2 - \sqrt{x^2 + 1}$ | $x \in \mathbb{R}$ |
| 6 | $f(x) = -e^{x/2} - x^2$ | $x \in \mathbb{R}$ |
| 7 | $f(x) = \log(1 + e^{-x})$ | $x \in \mathbb{R}$ |
| 8 | $f(x) = -\sqrt{1 + x^2}$ | $x \in \mathbb{R}$ |
| 9 | $f(x) = -\log(x^2 + 1)$ | $x \in \mathbb{R}$ |
| 10 | $f(x) = -x^6 - x^4 + x^3 - x + e^{-x^2}$ | $x \in \mathbb{R}$ |

Table 8: List of Continuous Concave Functions

## C APPENDIX: TEST DATASET IN DETAIL

Table 9,10,11 shows in detail the expression forms of the data set used in the experiment, as well as the sampling range and sampling number. Some specific presentation rules are described below

- The variables contained in the regression task are represented as $[x_1, x_2, ..., x_n]$.

- $U(a, b, c)$ signifies $c$ random points uniformly sampled between $a$ and $b$ for each input variable. Different random seeds are used for training and testing datasets.
- $E(a, b, c)$ indicates $c$ points evenly spaced between $a$ and $b$ for each input variable.

| Name | Expression | Dataset |
|---|---|---|
| Nguyen-1 | $x_1^3 + x_1^2 + x_1$ | $U(-1, 1, 20)$ |
| Nguyen-2 | $x_1^4 + x_1^3 + x_1^2 + x_1$ | $U(-1, 1, 20)$ |
| Nguyen-3 | $x_1^5 + x_1^4 + x_1^3 + x_1^2 + x_1$ | $U(-1, 1, 20)$ |
| Nguyen-4 | $x_1^6 + x_1^5 + x_1^4 + x_1^3 + x_1^2 + x_1$ | $U(-1, 1, 20)$ |
| Nguyen-5 | $\sin(x_1^2)\cos(x_1) - 1$ | $U(-1, 1, 20)$ |
| Nguyen-6 | $\sin(x_1) + \sin(x_1 + x_1^2)$ | $U(-1, 1, 20)$ |
| Nguyen-7 | $\log(x_1 + 1) + \log(x_1^2 + 1)$ | $U(0, 2, 20)$ |
| Nguyen-8 | $\sqrt{x_1}$ | $U(0, 4, 20)$ |
| Nguyen-9 | $\sin(x_1) + \sin(x_2^2)$ | $U(0, 1, 20)$ |
| Nguyen-10 | $2\sin(x_1)\cos(x_2)$ | $U(0, 1, 20)$ |
| Nguyen-11 | $x_1^{x_2}$ | $U(0, 1, 20)$ |
| Nguyen-12 | $x_1^4 - x_1^3 + \frac{1}{2}x_2^2 - x_2$ | $U(0, 1, 20)$ |
| Nguyen-2$'$ | $4x_1^4 + 3x_1^3 + 2x_1^2 + x_1$ | $U(-1, 1, 20)$ |
| Nguyen-5$'$ | $\sin(x_1^2)\cos(x_1) - 2$ | $U(-1, 1, 20)$ |
| Nguyen-8$'$ | $\sqrt[3]{x_1}$ | $U(0, 4, 20)$ |
| Nguyen-8$''$ | $\sqrt[3]{x_1^2}$ | $U(0, 4, 20)$ |
| Nguyen-1$^c$ | $3.39x_1^3 + 2.12x_1^2 + 1.78x$ | $U(-1, 1, 20)$ |
| Nguyen-5$^c$ | $\sin(x_1^2)\cos(x) - 0.75$ | $U(-1, 1, 20)$ |
| Nguyen-7$^c$ | $\log(x + 1.4) + \log(x_1^2 + 1.3)$ | $U(0, 2, 20)$ |
| Nguyen-8$^c$ | $\sqrt{1.23x}$ | $U(0, 4, 20)$ |
| Nguyen-10$^c$ | $\sin(1.5x)\cos(0.5x_2)$ | $U(0, 1, 20)$ |
| Korns-1 | $1.57 + 24.3 * x_1^4$ | $U(-1, 1, 20)$ |
| Korns-2 | $0.23 + 14.2\frac{(x_4+x_1)}{(3x_2)}$ | $U(-1, 1, 20)$ |
| Korns-3 | $4.9\frac{(x_2-x_1+\frac{x_1}{x_3})}{(3x_3))} - 5.41$ | $U(-1, 1, 20)$ |
| Korns-4 | $0.13\sin(x_1) - 2.3$ | $U(-1, 1, 20)$ |
| Korns-5 | $3 + 2.13\log(|x_5|)$ | $U(-1, 1, 20)$ |
| Korns-6 | $1.3 + 0.13\sqrt{|x_1|}$ | $U(-1, 1, 20)$ |
| Korns-7 | $2.1(1 - e^{-0.55x_1})$ | $U(-1, 1, 20)$ |
| Korns-8 | $6.87 + 11\sqrt{|7.23x_1 x_4 x_5|}$ | $U(-1, 1, 20)$ |
| Korns-9 | $12\sqrt{|4.2x_1 x_2 x_2|}$ | $U(-1, 1, 20)$ |
| Korns-10 | $0.81 + 24.3\frac{2x_1+3x_2^2}{4x_3^3+5x_4^4}$ | $U(-1, 1, 20)$ |
| Korns-11 | $6.87 + 11\cos(7.23x_1^3)$ | $U(-1, 1, 20)$ |
| Korns-12 | $2 - 2.1\cos(9.8x_1^3)\sin(1.3x_5)$ | $U(-1, 1, 20)$ |
| Korns-13 | $32.0 - 3.0\frac{tan(x_1)}{tan(x_2)}\frac{tan(x_3)}{tan(x_4)}$ | $U(-1, 1, 20)$ |
| Korns-14 | $22.0 - (4.2\cos(x_1) - \tan(x_2))\frac{tanh(x_3)}{sin(x_4)}$ | $U(-1, 1, 20)$ |
| Korns-15 | $12.0 - \frac{6.0tan(x_1)}{e^{x_2}}(log(x_3) - tan(x_4))))$ | $U(-1, 1, 20)$ |
| Jin-1 | $2.5x_1^4 - 1.3x_1^3 + 0.5x_2^2 - 1.7x_2$ | $U(-3, 3, 100)$ |
| Jin-2 | $8.0x_1^2 + 8.0x_2^3 - 15.0$ | $U(-3, 3, 100)$ |
| Jin-3 | $0.2x_1^3 + 0.5x_2^3 - 1.2x_2 - 0.5x_1$ | $U(-3, 3, 100)$ |
| Jin-4 | $1.5\exp x + 5.0\cos(x_2)$ | $U(-3, 3, 100)$ |
| Jin-5 | $6.0\sin(x_1)\cos(x_2)$ | $U(-3, 3, 100)$ |
| Jin-6 | $1.35x_1 x_2 + 5.5\sin((x_1 - 1.0)(x_2 - 1.0))$ | $U(-3, 3, 100)$ |

Table 9: Specific formula form and value range of the three data sets Nguyen, Korns, and Jin.

| Name | Expression | Dataset |
|------|-----------|---------|
| Neat-1 | $x_1^4 + x_1^3 + x_1^2 + x$ | $U(-1, 1, 20)$ |
| Neat-2 | $x_1^5 + x_1^4 + x_1^3 + x_1^2 + x$ | $U(-1, 1, 20)$ |
| Neat-3 | $\sin(x_1^2)\cos(x) - 1$ | $U(-1, 1, 20)$ |
| Neat-4 | $\log(x + 1) + \log(x_1^2 + 1)$ | $U(0, 2, 20)$ |
| Neat-5 | $2\sin(x)\cos(x_2)$ | $U(-1, 1, 100)$ |
| Neat-6 | $\sum_{k=1}^{x} \frac{1}{k}$ | $E(1, 50, 50)$ |
| Neat-7 | $2 - 2.1\cos(9.8x_1)\sin(1.3x_2)$ | $E(-50, 50, 10^5)$ |
| Neat-8 | $\frac{e^{-(x_1)^2}}{1.2+(x_2-2.5)^2}$ | $U(0.3, 4, 100)$ |
| Neat-9 | $\frac{1}{1+x_1^{-4}} + \frac{1}{1+x_2^{-4}}$ | $E(-5, 5, 21)$ |
| Keijzer-1 | $0.3x_1 sin(2\pi x_1)$ | $U(-1, 1, 20)$ |
| Keijzer-2 | $2.0x_1 sin(0.5\pi x_1)$ | $U(-1, 1, 20)$ |
| Keijzer-3 | $0.92x_1 sin(2.41\pi x_1)$ | $U(-1, 1, 20)$ |
| Keijzer-4 | $x_1^3 e^{-x_1} cos(x_1) sin(x_1) sin(x_1)^2 cos(x_1) - 1$ | $U(-1, 1, 20)$ |
| Keijzer-5 | $3 + 2.13 log(|x_5|)$ | $U(-1, 1, 20)$ |
| Keijzer-6 | $\frac{x_1(x_1+1)}{2}$ | $U(-1, 1, 20)$ |
| Keijzer-7 | $log(x_1)$ | $U(0, 1, 20)$ |
| Keijzer-8 | $\sqrt{(x_1)}$ | $U(0, 1, 20)$ |
| Keijzer-9 | $log(x_1 + \sqrt{x_1^2 + 1})$ | $U(-1, 1, 20)$ |
| Keijzer-10 | $x_1^{x_2}$ | $U(-1, 1, 20)$ |
| Keijzer-11 | $x_1 x_2 + sin((x_1 - 1)(x_2 - 1))$ | $U(-1, 1, 20)$ |
| Keijzer-12 | $x_1^4 - x_1^3 + \frac{x_2^2}{2} - x_2$ | $U(-1, 1, 20)$ |
| Keijzer-13 | $6 sin(x_1) cos(x_2)$ | $U(-1, 1, 20)$ |
| Keijzer-14 | $\frac{8}{2+x_1^2+x_2^2}$ | $U(-1, 1, 20)$ |
| Keijzer-15 | $\frac{x_1^3}{5} + \frac{x_2^3}{2} - x_2 - x_1$ | $U(-1, 1, 20)$ |
| Livermore-1 | $\frac{1}{3} + x_1 + sin(x_1^2))$ | $U(-3, 3, 100)$ |
| Livermore-2 | $sin(x_1^2) * cos(x1) - 2$ | $U(-3, 3, 100)$ |
| Livermore-3 | $sin(x_1^3) * cos(x_1^2)) - 1$ | $U(-3, 3, 100)$ |
| Livermore-4 | $log(x_1 + 1) + log(x_1^2 + 1) + log(x_1)$ | $U(-3, 3, 100)$ |
| Livermore-5 | $x_1^4 - x_1^3 + x_2^2 - x_2$ | $U(-3, 3, 100)$ |
| Livermore-6 | $4x_1^4 + 3x_1^3 + 2x_1^2 + x_1$ | $U(-3, 3, 100)$ |
| Livermore-7 | $\frac{(exp(x1) - exp(-x_1)}{2})$ | $U(-1, 1, 100)$ |
| Livermore-8 | $\frac{(exp(x1) + exp(-x1)}{3}$ | $U(-3, 3, 100)$ |
| Livermore-9 | $x_1^9 + x_1^8 + x_1^7 + x_1^6 + x_1^5 + x_1^4 + x_1^3 + x_1^2 + x_1$ | $U(-1, 1, 100)$ |
| Livermore-10 | $6 * sin(x_1) cos(x_2)$ | $U(-3, 3, 100)$ |
| Livermore-11 | $\frac{x_1^2 x_2^2}{(x_1+x_2)}$ | $U(-3, 3, 100)$ |
| Livermore-12 | $\frac{x_1^5}{x_2^3}$ | $U(-3, 3, 100)$ |
| Livermore-13 | $x_1^{\frac{1}{3}}$ | $U(-3, 3, 100)$ |
| Livermore-14 | $x_1^3 + x_1^2 + x_1 + sin(x_1) + sin(x_2^2)$ | $U(-1, 1, 100)$ |
| Livermore-15 | $x_1^{\frac{1}{5}}$ | $U(-3, 3, 100)$ |
| Livermore-16 | $x_1^{\frac{2}{3}}$ | $U(-3, 3, 100)$ |
| Livermore-17 | $4 sin(x_1) cos(x_2)$ | $U(-3, 3, 100)$ |
| Livermore-18 | $sin(x_1^2) * cos(x_1) - 5$ | $U(-3, 3, 100)$ |
| Livermore-19 | $x_1^5 + x_1^4 + x_1^2 + x_1$ | $U(-3, 3, 100)$ |
| Livermore-20 | $e^{(-x_1^2)}$ | $U(-3, 3, 100)$ |
| Livermore-21 | $x_1^8 + x_1^7 + x_1^6 + x_1^5 + x_1^4 + x_1^3 + x_1^2 + x_1$ | $U(-1, 1, 20)$ |
| Livermore-22 | $e^{(-0.5x_1^2)}$ | $U(-3, 3, 100)$ |

Table 10: Specific formula form and value range of the three data sets neat, Keijzer, and Livermore.

| Name | Expression | Dataset |
|------|-----------|---------|
| Vladislavleva-1 | $\frac{(e^{-(x1-1)^2})}{(1.2+(x2-2.5)^2))}$ | U$(-1, 1, 20)$ |
| Vladislavleva-2 | $e^{-x_1}x_1^3 cos(x_1)sin(x_1)(cos(x_1)sin(x_1)^2 - 1)$ | U$(-1, 1, 20)$ |
| Vladislavleva-3 | $e^{-x_1}x_1^3 cos(x_1)sin(x_1)(cos(x_1)sin(x_1)^2 - 1)(x_2 - 5)$ | U$(-1, 1, 20)$ |
| Vladislavleva-4 | $\frac{10}{5+(x1-3)^2+(x_2-3)^2+(x_3-3)^2+(x_4-3)^2+(x_5-3)^2}$ | U$(0, 2, 20)$ |
| Vladislavleva-5 | $30(x_1 - 1)\frac{x_3-1}{(x_1-10)}x_2^2$ | U$(-1, 1, 100)$ |
| Vladislavleva-6 | $6sin(x_1)cos(x_2)$ | E$(1, 50, 50)$ |
| Vladislavleva-7 | $2 - 2.1\cos(9.8x)\sin(1.3x_2)$ | E$(-50, 50, 10^5)$ |
| Vladislavleva-8 | $\frac{e^{-(x-1)^2}}{1.2+(x_2-2.5)^2}$ | U$(0.3, 4, 100)$ |
| Test-2 | $3.14x_1^2$ | U$(-1, 1, 20)$ |
| Const-Test-1 | $5x_1^2$ | U$(-1, 1, 20)$ |
| GrammarVAE-1 | $1/3 + x1 + sin(x_1^2))$ | U$(-1, 1, 20)$ |
| Sine | $sin(x_1) + sin(x_1 + x_1^2))$ | U$(-1, 1, 20)$ |
| Nonic | $x_1^9 + x_1^8 + x_1^7 + x_1^6 + x_1^5 + x_1^4 + x_1^3 + x_1^2 + x_1$ | U$(-1, 1, 100)$ |
| Pagie-1 | $\frac{1}{1+x_1^{-4}}+\frac{1}{1+x_2^{-4}}$ | E$(1, 50, 50)$ |
| Meier-3 | $\frac{x_1^2 x_2^2}{(x_1+x_2)}$ | E$(-50, 50, 10^5)$ |
| Meier-4 | $\frac{x_1^5}{x_2^3}$ | U$(0.3, 4, 100)$ |
| Poly-10 | $x_1 x_2 + x_3 x4 + x_5 x_6 + x_1 x_7 x_9 + x_3 x_6 x_{10}$ | E$(-1, 1, 100)$ |
| Constant-1 | $3.39 * x_1^3 + 2.12 * x_1^2 + 1.78 * x_1$ | $U(-4, 4, 100)$ |
| Constant-2 | $sin(x_1^2) * cos(x_1) - 0.75$ | $U(-4, 4, 100)$ |
| Constant-3 | $sin(1.5 * x_1) * cos(0.5 * x_2)$ | $U(0.1, 4, 100)$ |
| Constant-4 | $2.7 * x_1^{x_2}$ | $U(0.3, 4, 100)$ |
| Constant-5 | $sqrt(1.23 * x_1)$ | $U(0.1, 4, 100)$ |
| Constant-6 | $x_1^{0.426}$ | $U(0.0, 4, 100)$ |
| Constant-7 | $2 * sin(1.3 * x_1) * cos(x_2)$ | $U(-4, 4, 100)$ |
| Constant-8 | $log(x_1 + 1.4) + log(x1, 2 + 1.3)$ | $U(-4, 4, 100)$ |
| R1 | $\frac{(x_1+1)^3}{x_1^2-x_1+1)}$ | $U(-5, 5, 100)$ |
| R2 | $\frac{(x_1^2-3*x_1^2+1)}{x_1^2+1)}$ | $U(-4, 4, 100)$ |
| R3 | $\frac{x_1^6+x_1^5)}{(x_1^4+x_1^3+x_1^2+x1+1)}$ | $U(-4, 4, 100)$ |

Table 11: Specific formula form and value range of the three data sets Vladislavleva and others.

## D APPENDIX: CHATSR TESTS ON AIFEYNMAN DATASET.

In our study, we conducted an evaluation of our novel symbol regression algorithm, termed ChatSR, leveraging the AI Feynman dataset, which comprises a diverse array of problems spanning various subfields of physics and mathematics, including mechanics, thermodynamics, and electromagnetism. Originally, the dataset contained 100,000 data points; however, for a more rigorous assessment of ChatSR's efficacy, our analysis was deliberately confined to a subset of 100 data points. Through the application of ChatSR for symbol regression on these selected data points, we meticulously calculated the $R^2$ values to compare the algorithm's predictions against the true solutions.

The empirical results from our investigation unequivocally affirm that ChatSR possesses an exceptional ability to discern the underlying mathematical expressions from a constrained sample size. Notably, the $R^2$ values achieved were above 0.99 for a predominant portion of the equations, underscoring the algorithm's remarkable accuracy in fitting these expressions. These findings decisively position ChatSR as a potent tool for addressing complex problems within the domains of physics and mathematics. The broader implications of our study suggest that ChatSR holds considerable promise for a wide range of applications across different fields. Detailed experimental results are presented in Table 12 and Table 13.

| Feynman | Equation | $R^2$ |
|---|---|---|
| I.6.20a | $f = e^{-\theta^2/2}/\sqrt{2\pi}$ | 0.9989 |
| I.6.20 | $f = e^{-\frac{\theta^2}{2\sigma^2}}/\sqrt{2\pi\sigma^2}$ | 0.9973 |
| I.6.20b | $f = e^{-\frac{(\theta-\theta_1)^2}{2\sigma^2}}/\sqrt{2\pi\sigma^2}$ | 0.9421 |
| I.8.14 | $d = \sqrt{(x_2-x_1)^2 + (y_2-y_1)^2}$ | 0.9413 |
| I.9.18 | $F = \frac{Gm_1m_2}{(x_2-x_1)^2+(y_2-y_1)^2+(z_2-z_1)^2}$ | 0.9835 |
| I.10.7 | $F = \frac{Gm_1m_2}{(x_2-x_1)^2+(y_2-y_1)^2+(z_2-z_1)^2}$ | 0.9724 |
| I.11.19 | $A = x_1y_1 + x_2y_2 + x_3y_3$ | 0.9891 |
| I.12.1 | $F = \mu N_n$ | 0.9956 |
| I.12.2 | $F = \frac{q_1q_2}{4\pi\epsilon r^2}$ | 0.9999 |
| I.12.4 | $E_f = \frac{q_1}{4\pi\epsilon r^2}$ | 0.9939 |
| I.12.5 | $F = q_2 E_f$ | 0.9999 |
| I.12.11 | $F = \mathcal{Q}(E_f + Bv\sin\theta)$ | 0.9983 |
| I.13.4 | $K = \frac{1}{2}m(v^2 + u^2 + w^2)$ | 0.9913 |
| I.13.12 | $U = Gm_1m_2(\frac{1}{r_2} - \frac{1}{r_1})$ | 0.9859 |
| I.14.3 | $U = mgz$ | 1.0 |
| I.14.4 | $U = \frac{k_{spring}x^2}{2}$ | 0.9926 |
| I.15.3x | $x_1 = \frac{x-ut}{\sqrt{1-u^2/c^2}}$ | 0.9828 |
| I.15.3t | $t_1 = \frac{t-ux/c^2}{\sqrt{1-u^2/c^2}}$ | 0.9732 |
| I.15.10 | $p = \frac{m_0 v}{\sqrt{1-v^2/c^2}}$ | 0.9810 |
| I.16.6 | $v_1 = \frac{u+v}{1+uv/c^2}$ | 0.9984 |
| I.18.4 | $r = \frac{m_1r_1+m_2r_2}{m_1+m_2}$ | 0.9818 |
| I.18.12 | $\tau = rF\sin\theta$ | 0.9928 |
| I.18.16 | $L = mrv\sin\theta$ | 0.9993 |
| I.24.6 | $E = \frac{1}{4}m(\omega^2 + \omega_0^2)x^2$ | 0.9981 |
| I.25.13 | $V_e = \frac{q}{C}$ | 1.0 |
| I.26.2 | $\theta_1 = \arcsin(n\sin\theta_2)$ | 0.9992 |
| I.27.6 | $f_f = \frac{1}{\frac{1}{d_1} + \frac{n}{d_2}}$ | 0.9914 |
| I.29.4 | $k = \frac{\omega}{c}$ | 1.0 |
| I.29.16 | $x = \sqrt{x_1^2 + x_2^2 - 2x_1x_2\cos(\theta_1 - \theta_2)}$ | 0.9827 |
| I.30.3 | $I_* = I_{*0}\frac{\sin^2(n\theta/2)}{\sin^2(\theta/2)}$ | 0.9937 |
| I.30.5 | $\theta = \arcsin(\frac{\lambda}{nd})$ | 0.9917 |
| I.32.5 | $P = \frac{q^2a^2}{6\pi\epsilon c^3}$ | 0.9933 |
| I.32.17 | $P = (\frac{1}{2}\epsilon cE_f^2)(8\pi r^2/3)(\omega^4/(\omega^2 - \omega_0^2)^2)$ | 0.991 |
| I.34.8 | $\omega = \frac{qvB}{p}$ | 0.9999 |
| I.34.10 | $\omega = \frac{\omega_0}{1-v/c}$ | 0.9913 |
| I.34.14 | $\omega = \frac{1+v/c}{\sqrt{1-v^2/c^2}}\omega_0$ | 0.9918 |
| I.34.27 | $E = \hbar\omega$ | 0.9972 |
| I.37.4 | $I_* = I_1 + I_2 + 2\sqrt{I_1I_2}\cos\delta$ | 0.9827 |
| I.38.12 | $r = \frac{4\pi\epsilon\hbar^2}{mq^2}$ | 0.9983 |
| I.39.10 | $E = \frac{3}{2}p_F V$ | 0.9965 |
| I.39.11 | $E = \frac{1}{\gamma-1}p_F V$ | 0.9792 |
| I.39.22 | $P_F = \frac{nk_bT}{V}$ | 0.9935 |
| I.40.1 | $n = n_0 e^{-\frac{mgx}{k_bT}}$ | 0.9799 |
| I.41.16 | $L_{rad} = \frac{\hbar\omega^3}{\pi^2c^2(e^{\frac{\hbar\omega}{k_bT}}-1)}$ | 0.9983 |
| I.43.16 | $v = \frac{\mu_{drift}qV_e}{d}$ | 0.9981 |
| I.43.31 | $D = \mu_e k_bT$ | 1.0 |
| I.43.43 | $\kappa = \frac{1}{\gamma-1}\frac{k_bv}{A}$ | 0.9347 |
| I.44.4 | $E = nk_bT\ln(\frac{V_2}{V_1})$ | 0.9024 |
| I.47.23 | $c = \sqrt{\frac{\gamma pr}{\rho}}$ | 0.9724 |
| I.48.20 | $E = \frac{mc^2}{\sqrt{1-v^2/c^2}}$ | 0.8902 |
| I.50.26 | $x = x_1[\cos(\omega t) + \alpha\,cos(\omega t)^2]$ | 0.9999 |

Table 12: Tested Feynman Equations, part 1.

| Feynman | Equation | $R^2$ |
|---|---|---|
| II.2.42 | $P = \frac{\kappa(T_2 - T_1)A}{d}$ | 0.8824 |
| II.3.24 | $F_E = \frac{P}{4\pi r^2}$ | 0.9820 |
| II.4.23 | $V_e = \frac{q}{4\pi\epsilon r}$ | 0.9888 |
| II.6.11 | $V_e = \frac{1}{4\pi\epsilon}\frac{p_d \cos\theta}{r^2}$ | 0.9837 |
| II.6.15a | $E_f = \frac{3}{4\pi\epsilon}\frac{p_d z}{r^5}\sqrt{x^2 + y^2}$ | 0.9235 |
| II.6.15b | $E_f = \frac{3}{4\pi\epsilon}\frac{p_d}{r^3}\cos\theta\sin\theta$ | 0.9928 |
| II.8.7 | $E = \frac{3}{5}\frac{q^2}{4\pi\epsilon d}$ | 0.9827 |
| II.8.31 | $E_{den} = \frac{\epsilon E_f^2}{2}$ | 0.9999 |
| II.10.9 | $E_f = \frac{\sigma_{den}}{\epsilon}\frac{1}{1+\chi}$ | 0.9933 |
| II.11.3 | $x = \frac{qE_f}{m(\omega_0^2 - \omega^2)}$ | 0.9918 |
| II.11.7 | $n = n_0(1 + \frac{p_d E_f \cos\theta}{k_b T})$ | 0.8927 |
| II.11.20 | $P_* = \frac{n_\rho p_d^2 E_f}{3 k_b T}$ | 0.8355 |
| II.11.27 | $P_* = \frac{n\alpha}{1 - n\alpha/3}\epsilon E_f$ | 0.9925 |
| II.11.28 | $\theta = 1 + \frac{n\alpha}{1 - (n\alpha/3)}$ | 0.9992 |
| II.13.17 | $B = \frac{1}{4\pi\epsilon c^2}\frac{2I}{r}$ | 0.9993 |
| II.13.23 | $\rho_c = \frac{\rho_{c_0}}{\sqrt{1 - v^2/c^2}}$ | 0.9902 |
| II.13.34 | $j = \frac{\rho_{c_0} v}{\sqrt{1 - v^2/c^2}}$ | 0.9827 |
| II.15.4 | $E = -\mu_M B \cos\theta$ | 0.9997 |
| II.15.5 | $E = -p_d E_f \cos\theta$ | 0.9973 |
| II.21.32 | $V_e = \frac{q}{4\pi\epsilon r(1 - v/c)}$ | 0.9910 |
| II.24.17 | $k = \sqrt{\frac{\omega^2}{c^2} - \frac{\pi^2}{d^2}}$ | 0.9837 |
| II.27.16 | $F_E = \epsilon c E_f^2$ | 0.9935 |
| II.27.18 | $E_{den} = \epsilon E_f^2$ | 0.9972 |
| II.34.2a | $I = \frac{qv}{2\pi r}$ | 0.9980 |
| II.34.2 | $\mu_M = \frac{qvr}{2}$ | 0.9903 |
| II.34.11 | $\omega = \frac{g_- q B}{2m}$ | 0.9937 |
| II.34.29a | $\mu_M = \frac{qh}{4\pi m}$ | 0.9938 |
| II.34.29b | $E = \frac{g_- \mu_M B J_z}{\hbar}$ | 0.9037 |
| II.35.18 | $n = \frac{n_0}{\exp(\mu_m B/(k_b T)) + \exp(-\mu_m B/(k_b T))}$ | 0.9738 |
| II.35.21 | $M = n_\rho \mu_M \tanh(\frac{\mu_M B}{k_b T})$ | 0.8537 |
| II.36.38 | $f = \frac{\mu_m B}{k_b T} + \frac{\mu_m \alpha M}{\epsilon c^2 k_b T}$ | 0.9928 |
| II.37.1 | $E = \mu_M(1 + \chi)B$ | 0.9999 |
| II.38.3 | $F = \frac{Y A x}{d}$ | 0.9985 |
| II.38.14 | $\mu_S = \frac{Y}{2(1 + \sigma)}$ | 0.9988 |
| III.4.32 | $n = \frac{1}{e^{\frac{\hbar\omega}{k_b T}} - 1}$ | 0.9903 |
| III.4.33 | $E = \frac{\hbar\omega}{e^{\frac{\hbar\omega}{k_b T}} - 1}$ | 0.9984 |
| III.7.38 | $\omega = \frac{2\mu_M B}{\hbar}$ | 0.9973 |
| III.8.54 | $p_\gamma = \sin(\frac{Et}{\hbar})^2$ | 0.9973 |
| III.9.52 | $p_\gamma = \frac{p_d E_f t}{\hbar}\frac{\sin((\omega - \omega_0)t/2)^2}{((\omega - \omega_0)t/2)^2}$ | 0.8036 |
| III.10.19 | $E = \mu_M \sqrt{B_x^2 + B_y^2 + B_z^2}$ | 0.9935 |
| III.12.43 | $L = n\hbar$ | 0.9999 |
| III.13.18 | $v = \frac{2E d^2 k}{\hbar}$ | 0.9935 |
| III.14.14 | $I = I_0(e^{\frac{qV_e}{k_b T}} - 1)$ | 0.9927 |
| III.15.12 | $E = 2U(1 - \cos(kd))$ | 0.9993 |
| III.15.14 | $m = \frac{\hbar^2}{2E d^2}$ | 0.9927 |
| III.15.27 | $k = \frac{2\pi\alpha}{nd}$ | 0.9999 |
| III.17.37 | $f = \beta(1 + \alpha\cos\theta)$ | 0.9938 |
| III.19.51 | $E = \frac{-m q^4}{2(4\pi\epsilon)^2 \hbar^2}\frac{1}{n^2}$ | 0.9974 |
| III.21.20 | $j = \frac{-\rho_{c_0} q A_{vec}}{m}$ | 0.8668 |

Table 13: Tested Feynman Equations, part 2.

