# OpenReview forum: "ChatSR: Conversational Symbolic Regression"
_ICLR.cc/2025/Conference — ICLR 2025 Conference Withdrawn Submission_

### Official Review · Reviewer_1tvd · 2024-10-22

**Soundness:** 2
**Presentation:** 2
**Contribution:** 2
**Rating:** 3
**Confidence:** 3

**Summary:**

This paper introduces ChatSR, a conversational symbolic regression method based on MLLM. The key innovation is the ability to generate mathematical expressions that fit observed data while meeting specific requirements described in natural language. The method combines a SetTransformer for data feature extraction, an LLM for understanding natural language instructions and generating expressions, and a projection layer to align data and text features. ChatSR demonstrates state-of-the-art performance in fitting accuracy and exhibits zero-shot capabilities in understanding and applying previously unseen requirements.

**Strengths:**

ChatSR presents a novel approach to symbolic regression by integrating natural language understanding with expression generation. This allows for intuitive user interaction and incorporation of domain knowledge.
The method performs strongly, outperforming state-of-the-art baselines in fitting accuracy. Its ability to handle complex requirements and exhibit zero-shot learning is particularly impressive.
This work opens up new scientific discovery and data analysis possibilities by making symbolic regression more accessible and flexible.

**Weaknesses:**

The introduction lacks proper citations, which are crucial for contextualizing the work and acknowledging prior research
Proper use of \citep and \citet commands with appropriate spacing is needed throughout the paper.
The related work section is excessively long, potentially overshadowing the paper's own contributions.
The table captions are positioned below the tables, which doesn't adhere to the ICLR template requirements.
Figure 4 lacks clarity in its legend, text, and content. Improving the visual presentation would enhance the reader's understanding of the results.
The paper provides a limited analysis of the experimental results.
In Figure 1, "Larger Language Model" should likely be "Large Language Model."
The paper lacks discussion on computational requirements and other important aspects of the experimental setup. Including this information would provide valuable context for the method's practical applicability and reproducibility.

**Questions:**

see weakness

---

### Official Review · Reviewer_Bx5x · 2024-10-31

**Soundness:** 2
**Presentation:** 2
**Contribution:** 1
**Rating:** 3
**Confidence:** 4

**Summary:**

The paper introduces ChatSR, a symbolic regression method leveraging multimodal large language models (MLLMs) to enable conversational interactions. Unlike traditional approaches that rely solely on observed data, ChatSR allows users to describe desired properties of the target expressions in natural language, such as periodicity or specific symbols like "sin." Through a feature extractor and projection layer, the method maps data features to word features, facilitating this natural language input. ChatSR achieves state-of-the-art performance across various datasets and exhibits strong zero-shot capabilities, generating valid expressions even for requirements unseen during training.

**Strengths:**

1. Users can specify requirements in natural language without needing code modifications, lowering the barrier to entry for symbolic regression.

2. The LLM interprets prior knowledge from natural language instructions, generating higher-quality expressions as a result.

3. Integrating MLLMs for symbolic regression (SR) is a novel approach, creating new opportunities for user-friendly applications.

4. The model’s capability to generate expressions for previously unseen properties highlights the potential of LLMs in this domain.

**Weaknesses:**

1. The paper lacks a strong motivation. It seems to primarily revise existing Transformer-based symbolic regression (SR) methods by simply replacing the decoder layers with decoder-only LLM layers.

2. Although incorporating prior knowledge enhances performance, it restricts generalization. The model’s success relies heavily on the presence of prior knowledge in the prompts.

3. Training the model—especially due to the feature alignment process and use of LLMs—requires substantial computational resources, a limitation that is insufficiently addressed in the paper.

4. The writing quality is poor and requires extensive proofreading to improve clarity and readability. There are numerous grammatical errors (notably in Section 2.2) and citation formatting issues (e.g., “RSRMXu et al.,” and “Double Q learningHasselt (2010)”).

5. The paper’s structure is confusing, with frequent grammatical errors, improper citation formatting, and some unclear figures and experimental results, making it challenging to understand the authors' intended message.

**Questions:**

(1) In Section 1, this paper defines $X \in R^{n \times d}$ and $y \in R^n$. Under this definition, $\{X,y\}\in R^{n \times (d + 1)}$. However, in Section 3.4.1, $D=\{X,y\}\in R^{n \times d}$ is used. Please clarify.

(2) Figure 2 in this paper bears a strong resemblance to Figure 1 in [1].

(3) In Section 3.4, this paper states that Vicuna is used as the LLM, but in the last paragraph of Section 3.5, LLaVA is mentioned instead. Please clarify which LLM was used in your experiments and provide more details.

(4) The definition of $X_{instruct}^t$ is ambiguous when $t = 1$ in equation (1). This leads to confusion, as two $x_D$ terms are included in equation (2).

(5) In equation (2), $X_{instruct,<i}$ should likely be $X_{instruct}$. Please verify.

(6) The MMSR results reported in Table 1 of this paper for expression complexity (Nodes) are consistent with those in [2], but the $R^2$ results are significantly lower. Please clarify.

(7) While Section 2 contains numerous references, but no citations are present in Section 1. What is the rationale for this omission?

(8) Does the training data actually match what is shown in Figure 3? (The examples contain multiple grammatical errors)

(9)  Section 3.5 mentions the use of multi-turn dialogue data, but the examples in Figure 2 show no clear relationship between the turns. Why not use single-turn data for training?

(10)  Table 1 reports an Average $R^2$ of 0.9820 for MMSR [2], while MMSR [2] reports 0.9934, significantly outperforming ChatSR. Given that the datasets and evaluation metrics should be consistent, what accounts for this discrepancy?


References:

[1] Kazem Meidani, Parshin Shojaee, Chandan K Reddy, and Amir Barati Farimani. Snip: Bridging math

ematical symbolic and numeric realms with unified pre-training. The Twelfth International Conference on Learning Representations, 2024.

[2] Yanjie Li, Jingyi Liu, Min Wu, Lina Yu, Weijun Li, Xin Ning, Wenqiang Li, Meilan Hao, Yusong Deng,

and Shu Wei. Mmsr: Symbolic regression is a multi-modal information fusion task. Information Fusion, 2024.

---

### Official Review · Reviewer_dmb6 · 2024-11-03

**Soundness:** 2
**Presentation:** 2
**Contribution:** 2
**Rating:** 3
**Confidence:** 3

**Summary:**

This work proposes ChatSR, a conversational symbolic regression approach that enables finding mathematical expressions from observed data while following natural language instructions and constraints. The main contribution is its ability to understand and incorporate specific constraints (like symmetry, periodicity, or inclusion of certain mathematical symbols) through natural language instructions, making it more accessible to non-computer science users. The method leverages a multimodal large language model architecture that combines SetTransformer for data feature extraction with a language model for expression generation, trained on 15M synthetic question-answer pairs of data and language instructions. Experimental results show that ChatSR outperforms state-of-the-art baselines in both expression accuracy (R² score) and expression simplicity (lower node count), while also demonstrating strong zero-shot capabilities for properties not seen during training.

**Strengths:**

- The comprehensive background explanations of symbolic regression tasks make the paper accessible and easy to understand

- The high-level figures provide clear visual aids that effectively guide readers through the methodology and concepts

**Weaknesses:**

Hard to understand where the performance gain came from:
As far as I understood, the proposed approach is to generate a dataset consisting of QA pairs where questions contain some constraints and train a model with the dataset. However, the datasets used in Table 1 do not require the ability to follow natural language instructions. Despite the performance gain, it seems that the effectiveness of their approach is not fully supported by these results (Table 1).

Unreliable results of the baseline:
The results of MMST in Table 1 are extremely low compared to the results in the original paper. For example, the reported R² score of MMSR was 0.9937±0.004, but in Table 1, the score is 0.9037±0.004. If the results are reproduced by the authors, they should carefully check the implementation of MMSR to provide a reliable comparison. Besides, the performance of NeSymRes is identical to the reported performance in the MMSR paper and it seems that the authors did not implement NeSymRes by themselves but used the reported scores.

Lack of analysis on the quality of the synthesized data:
While I agree with the direction of this work which aims to train multi-modal language models to follow instructions, more in-depth analyses are required to better understand the quality of the generated data and the effectiveness of the generation pipeline. For example, the authors might analyze the complexity of the constraints in the input questions.

The analysis (Table 2) is quite trivial:
Providing prior knowledge will naturally act as a hint for the regression of the solution function. In Line 480, the authors provide an example of prior knowledge: "For example, for Nguyen-5 sin(x²)cos(x) − 1, we'll ask it to generate an expression that contains the symbols sin and cos". However, it seems very trivial that a model would perform better with more information about the solution.

**Questions:**

- Is the conversation a multi-turn dialog? In Line 356, the authors mention collecting multi-turn QA pairs as the dataset. However, there is no multi-turn example in Figure 3. Additionally, please provide statistics of the dataset (e.g., average number of turns and diversity of the questions).

- If the goal of the approach is to improve the ability to conform to constraints in the input question, it would be better to provide experimental results on how well ChatSR meets the provided constraints.

- This work has significant overlap with constrained generation. For example, previous work in constrained generation proposed generation algorithms to ensure adherence to constraints such as including specific keywords in summaries or limiting completion length. It would be beneficial to add related work in this area.

- Please use proper citation commands. The authors only use \citet command which makes it difficult for readers to follow the main content.

---

### Official Review · Reviewer_mcej · 2024-11-04

**Soundness:** 3
**Presentation:** 2
**Contribution:** 3
**Rating:** 6
**Confidence:** 3

**Summary:**

This paper introduces a symbolic regression method based on multimodal LLMs. Compared with previous methods, the method allows users to enter their prior/assumptions in natural language for expressions, which removes the need for adding constraints to the search process in RL methods or code changes.

**Strengths:**

+ The authors present a well-designed method to construct large-scale, high-quality synthesized QA data. Human-designed constraints are applied to validate candidate expressions.
+ The method is evaluated on 13 datasets, and the experimental results demonstrate the effectiveness of the proposed method.

**Weaknesses:**

+ The types of symbols considered are relatively limited (Line 212). Please elaborate on the comparisons of the types of symbols explored in this work and previous work. I'm also wondering if a more fine-grained, detailed description of the properties of expression may be helpful to increase the diversity of the instructions in data.
+ The writing would benefit from another round of proofreading and more details. Please check the suggestions in the question section.

**Questions:**

Suggestions/questions:
+ Please use \citep instead of \cite for better readability
+ The abbreviation such as "MSDB" in the related work is not so friendly for readers who are not familiar with the representative methods in this field.
+ Line 201: [X, Y]->[X, y]?
+ There is a missing space before "by" in Line 215
+ Please briefly the relationship between Vicuna (Line 337) and LLaVA (Line 383) to improve readability.
+ Please list the scale of the pretraining/post-training data used by other methods in the experiment section.

---

### Note · Authors · 2025-01-21

**Comment:**

Thank you very much for your careful review of our work. I wish you a happy life！

**Withdrawal Confirmation:**

I have read and agree with the venue's withdrawal policy on behalf of myself and my co-authors.